# Migration of Styrene in Yogurt and Dairy Products Packaged in Polystyrene: Results from Market Samples

**DOI:** 10.3390/foods11142120

**Published:** 2022-07-17

**Authors:** Valeria Guazzotti, Veronika Hendrich, Anita Gruner, Dominik Fiedler, Angela Störmer, Frank Welle

**Affiliations:** Fraunhofer Institute for Process Engineering and Packaging, IVV. Giggenhauser Straße 35, 85354 Freising, Germany; veronika.hendrich@ivv.fraunhofer.de (V.H.); anita.gruner@ivv.fraunhofer.de (A.G.); dominik.fiedler@ivv.fraunhofer.de (D.F.); angela.stoermer@ivv.fraunhofer.de (A.S.); frank.welle@ivv.fraunhofer.de (F.W.)

**Keywords:** styrene, migration, polystyrene, food contact materials, packaged foodstuffs, yogurt, cream, dairy products

## Abstract

The European Food Safety Authority is re-evaluating styrene for assessing the safety of food contact materials (FCM) such as polystyrene (PS) and started a systematic review of the data on migration levels in food. A restriction for styrene is expected in the near future. The main food contact application of PS is dairy packaging, mainly at refrigerated storage. In this study, seventeen dairy products packed in PS taken from the Italian and German markets were investigated. Styrene concentrations in the refrigerated dairy products (yogurt, cream) ranged from 5 to 30 µg/kg at the best before date, while in single serving portions of coffee creamer, which were stored at room temperature until the best before date of approx. 190 days, 401 µg/kg were measured. Among several parameters, the ratio between the surface contact area of the package and the quantity of the food packed, the time/temperature conditions of production/filling and storage of the products were identified as the main factors influencing styrene migration into food under realistic conditions. Yogurts fermented in the pots for approximately 8 h at 40–50 °C showed higher styrene levels than those fermented in an incubator and filled at 20 °C. The fat content might influence the styrene level but the effect, if any, was too small in relation to the variability of other parameters. Levels of styrene migrating into 50% ethanol food simulant under standardized condition (10 days/40 °C) were found to be much higher than levels in refrigerated foods. This raises the question as to whether compliance testing for PS plastics should be adapted taking into consideration the correlation between migration testing by laboratory simulations and migration into real food.

## 1. Introduction

Yogurt and other dairy products, such as cream, sour cream and milk desserts, are perishable foods because of the complex composition of their matrix and the susceptibility to microbial contamination. For the dairy industry, the selection of an appropriate packaging material, which maintains product safety and quality characteristics, plays a significant role. As for all packaged food products, the package must provide a safe, attractive, functional and cost-effective means for protecting the product throughout distribution and merchandising, for presenting it to the consumer, and for enabling easy consumption. Factors affecting the environmental impact of packaging must also be taken into consideration, and these are likely to become even more important in the future [1].

In Europe, yogurt and dairy products are currently predominantly packed in polypropylene (PP) or polystyrene (PS) cups or pots. Between the two polymer types, polystyrene packaging seems to be preferable for limiting aroma compound losses and for avoiding the development of odor defects [2]. Commercial yogurts can be produced in two different forms: set-style or stirred. The set type (also known as Balkan or Swiss yogurt) has a thick texture and sour taste, for its production the inoculated milk is filled when it is warm (approx. 40 °C) and directly fermented in the retail packaging for 8 h or more, until the desired acidity (pH < 4.4) is attained. Stirred yogurt (also known as European-style yogurt) is creamy and smooth, most often added with fruits and flavors to the final yogurt. It is produced by incubating the milk and the yogurt culture in a large tank, followed by stirring, cooling (to approx. 20 °C) and finally packed in individual cups [3]. Both are stored under refrigerated conditions.

PS is the material of choice for packaging milk products fermented in the retail packaging, such as “set” yogurt and sour cream. Indeed, PS is rather permeable to oxygen, carbon dioxide and water vapor, therefore it avoids bloated packaging, allowing adequate gaseous exchange, which is necessary during the fermentation process conducted by starter cultures (mainly probiotic lactic acid bacteria) in these dairy products. However, also for milk products not fermented in the packaging but with living cultures, such as “stirred” yogurt, the use of PS is preferred because its gaseous permeability enhances the survival and activity rate of the added microorganisms during refrigerated storage [4].

The most popular material in current use for packaging of fresh yogurt products is high impact polystyrene (HIPS) in the form of small cups, pots or larger tubs with either an aluminum foil/plastic laminate or a paper/plastic laminate heat-seal lid or closure [1]. HIPS plastics are produced by blending styrene-butadiene co-polymer as impact modifier and crystal PS (or general-purpose polystyrene, GPPS). HIPS can be further blended with GPPS at the stage of production of the thermoforming film; the ratios being selected to achieve the required balance of physical properties for the different forms of packaging and the conversion process. Containers may be produced in form–fill–seal packaging machines (where the completely packaged goods are produced in-line, starting from polymer granules, extrusion, thermoforming, filling and sealing) or be delivered in preformed packaging from packaging materials suppliers. It is normal to add pigments such as titan dioxide (TiO_2_) to the HIPS in order to improve the appearance of the package and to provide some barrier to light. This also helps in heating and softening the HIPS sheet for thermoforming when radiant heating is used [5]. White pigments are most commonly used, but other colors are also popular.

In Europe, plastics used in applications with food contact is subject to an in-depth safety assessment according to the Plastics Regulation (EC) No. 10/2011, which establishes a Union List of substances that are permitted for use in the manufacture of plastic FCMs, overall and specific restrictions (migration limits) and lays down detailed rules for the migration testing conditions to be applied to determine compliance. Moreover, according to the Framework Regulation (EC) No. 1935/2004, in order to retain the quality of the food product, all food packaging must not induce a detectable change in the taste, aroma, color or consistency of the packed food. The deterioration of the organoleptic characteristics of the food is probably the most sensitive criteria for PS packaging to consumer acceptance and regulatory concern [6] especially related to styrene migration.

Both manufacturers/suppliers of food packaging and food companies need to comply with the EU’s food contact regulation. Businesses are required to provide a “declaration of compliance” at all stages of the supply chain, apart from the retail stage. Compliance work shall be conducted as high up the supply chain as possible; usually migration testing is performed on finished food contact materials and articles.

The substances which can migrate from HIPS are mainly residual monomers (i.e., styrene and butadiene), oligomers and additives. Regarding styrene, it is an authorized monomer, listed in the European Food Packaging Legislation without a specific restriction, except for the requirement of sensorial inertness as styrene is a potent sensory active compound. Recently, the European Food Safety Authority’s (EFSA) classified styrene in the high priority group of substances needing re-evaluation among those that are authorized for use in food contact material without a specific migration limit [7]. Additionally, on the basis of human occupational and animal studies, both concerning exposure to styrene by inhalation, the International Agency for Research on Cancer (IARC) revised in 2019 the classification of styrene from “possibly carcinogenic to humans” (2B) to “probably carcinogenic to humans” (2A) [8]. Following this, in 2020, the EFSA published an assessment on the safety of styrene for use in plastic food contact materials [9], stating that the IARC conclusions cannot be “directly applied” to the evaluation of risks for consumers from oral exposure to styrene. However, it added that, “a concern for genotoxicity associated with oral exposure to styrene cannot be excluded” and a systematic review of the data is further required for assessing the safety of styrene and styrene oligomers presence in food contact materials.

The migration of styrene from PS has been studied in various foods and food simulants (test media which shall mimic real foodstuff, according to the European Plastics Regulation (EC) No. 10/2011) and it was found to be strongly dependent on its residual level content in the polymer, the fat content of the food/type of simulant as well as the temperature and time storage conditions [10,11,12,13,14,15,16,17,18,19,20,21]. An increase in styrene migration was found with an increasing amount of HIPS in the PS blends [22]. In a further study, the addition of nanoparticles (clay and ZnO) to polystyrene matrix was found to reduce the transfer of styrene monomer from polystyrene into the food simulant [23]. Moreover, the sampling method was reported to have an effect on the migration extent of monomer styrene; PS test specimens with cut edges on all-sided contacting test media led to the overestimation of transfer rates, due to damage at the edge of the test plaques [24,25]. Recently, it was demonstrated that migration tests on several styrenic plastics under the standardized conditions laid down in the Plastics Regulation (EC) No. 10/2011 (10 days/40 °C or 10 days/60 °C) with the food simulants 50% ethanol (simulant for dairy products), isooctane or 95% ethanol (alternative simulants for fatty foods), cause a strong swelling of these polymers, which does not take place under actual use conditions with real foods, including milk and cream [26,27,28]. Swelling results in significantly higher diffusion coefficients, leading to increased migration values [29]. Therefore, migration tests applied within laboratory simulations to evaluate regulatory compliance appear to be often too severe and not representative for the majority of the real food packaging application of PS, which are mostly refrigerated dairy products (such as yogurt and creams) with single side contact and a shelf life of up to approx. 40 days. On the other hand, only limited data have been published on the levels of styrene in yogurt and dairy products packed in PS directly taken from the market (retail surveys). Reported concentrations of styrene in the available literature vary considerably and are sometimes controversial (Table 1).

Comprehensive information/description of the analyzed products (packaging and food content) which were sampled at retail level is frequently missing in the retrieved studies. Particularly difficult is the evaluation/distinction of the source of styrene; since measured concentrations could arise not only from migration of the monomer from the food packaging (styrene plastics) but also through natural occurrence in food. Indeed, styrene has been identified as natural constituent/fermentation product or as environmental contaminant in a wide variety of foods without contact with PS packaging. Levels of styrene in food vary widely, from traces or intermediate levels in fruits, cheese, coffee, beef and oat to high levels in some moldy cheese and stored milled olives and very high levels (up to 40 mg/kg) in cinnamon [42,44,45].

In view of the above, the aim of the present study was to assess the level of styrene in yogurt and dairy products packed in polystyrene and to identify the possible main factors influencing migration of this monomer in real packed foods taken from the market. A limited number of samples collected from the Italian and German markets were investigated (seventeen in total); several characteristics for each product were reported, such as the residual levels of styrene, the area/weight and the food contact area of the packaging; the fat content, the net weight and a description of the food as well as of the production process and of the storage time/temperature conditions until analysis. This allowed for an evaluation and comparison of the results obtained in the present study with the literature data on styrene migration into food.

## 2. Materials and Methods

### 2.1. Sampling

Seventeen dairy products packed in PS were purchased during 2021 from different Italian and German supermarkets. To avoid matrix interferences or styrene contamination from other origins as the PS packaging, only plain (natural) products, without addition of flavorings, spices, herbs, fruit, chocolate or other preparations, were selected. Products packed in PS of different brands were targeted, looking at the bottom of the pots for their labelling with the recycling code for PS = 6. Three stirred yogurts with fat content ranging from 0.1% to 4.4% were sampled in Italy. Three stirred yogurts with 3.5% fat content and four types of set yogurts with a fat content of 3.5–3.9%, as well the other dairy products, were sampled in Germany. The other dairy products were one sample of whipping cream (Schlagsahne) with 30% fat (fluid, not homogenized, with cream on top), one sample of solid sour cream (Schmand) with 24% fat, four samples of solid/semi-solid sour cream (Sauerrahm or saure Sahne) with 10% fat and one sample of UHT coffee creamers (Kaffesahne) with 10% fat. All of the samples (except for the coffee creamers) were refrigerated products and according to the information of the producers have an average shelf life of 40 days at 5 °C. The coffee creamers were UHT products stored at room temperature and have an average shelf life of 192 days. In Table 2, the food description and the main characteristics of the investigated products (packaging and content) are summarized.

### 2.2. Determination of the Migration Levels of Styrene in Yogurt/Dairy Products from the Market by Purge & Trap Gas Chromatography Analysis

After buying in the supermarket, the samples were stored cooled in a refrigerator at 5 °C (except for the coffee creamers that are UHT products and not refrigerated at retail level which were stored at room temperature). At the best before date (BBD) of each sample (as reported on each product from the producer), the pots were opened, each yogurt/dairy product was stirred, and 1 g was weighted in glass vials) which were frozen until analysis (triplicate per sample). Some of the samples were additionally investigated approx. 20 days before and 15 days after the indicated BBD. In the case of the multipack, analyses at different dates were performed using the different portion; in the case of single selling unit, to perform analyses at different dates more products were bought.

For analysis, 5 mL of freshly boiled and cooled down water (HPLC grade) and 10 µL of an internal standard solution 4.52 ppm of m-xylene were added to each sample after defrosting and shaken for approximately 2 h using a flat bad shaker. Styrene was determined by purge & trap gas chromatography equipment (Thermo Scientific Trace GC, with VSP Autosampler, IMT, Moosbach, Germany) coupled with both mass spectrometry and flame ionization detector (P&T/GC/MS + FID). The headspace of the sample vials was heated at 40 °C and purged with helium (for 20 min at a flow rate of 20 mL/min). A Peltier water trap and a cooled (−35 °C) Tenax^®^ trap were used to remove humidity, and to adsorb and concentrate the volatile compounds which after desorption at 240 °C for 7 min were introduced by flash heating (800 °C/s) into the gas chromatograph for analysis. The column used was a Restek RXI 624 MS Sil with 60 m length, 0.32 mm inner diameter and 1.8 µm film. The temperature program was 40 °C for 6 min, then heating with 5 °C/min to 90 °C, then heating again with 10 °C/min to 260 °C for 15 min. The mass spectrometer was set in full scan mode (35–350 amu) at 0.2 scans/s. A splitter separated the eluting substances 1:1 to MS and FID. Quantification was obtained using the FID response, MS was used to confirm the identification of peaks.

External calibration and determination of the detection limits for styrene were carried out using “reference food matrices”. These were bought in supermarkets and were not packed in PS. They were chosen for their similarity with the investigated yogurt and dairy products packed in PS and were: one stirred and one set yogurt with 3.5% fat, one sour cream with 10% fat and one fat sour cream with 24% fat, one whipping cream with 30% fat and one coffee cream with 10% fat. Standard solutions diluted from a stock solution prepared in ethanol (over a range of 5–9 concentration levels depending on the food matrix) of styrene and of m-xylene as internal standard were added to each reference product in twice and after homogenization, 5 g of each of the spiked food was analyzed to obtain a standard calibration curve. To evaluate recovery and possible losses of styrene during freezing/defrosting, control standards (spiked reference foods not packed in PS) stored with the investigated samples in the freezer, as well as freshly prepared ones, were analyzed. All of the chemicals used in the experiments were of analytical grade and purchased by Merck (Darmstadt, Germany). All of the measurements were performed in triplicate. The limit of detection was defined as three times, and the limit of quantification as ten times, the signal-to-noise ratio.

### 2.3. Determination of the Residual Levels of Styrene Monomer in the PS Packaging from the Market

The concentrations of the styrene monomer were determined in the PS packaging (after emptying the food contained at its best before date and gently washing with water) by extraction with the solvent acetone and subsequent analysis by gas chromatography with flame ionization detection (GC-FID) as described in a previous research work [27]. In order to avoid interferences, only parts of the packaging without labels or printing (if present) were analyzed.

### 2.4. Semi-Quantification of TiO_2_ as Ti in the PS Packaging from the Market by ED XRF Analysis

An additional analysis by energy dispersive X-ray fluorescence spectrometry (ED XRF) was performed on seven PS yogurt pots sampled from the market (items: 1 to 7 Germany). For the analysis, circles of approx. 38 mm diameter were cut out from a single pot and then stacked. The obtained sample pack of 1.5–3 g PS sample (depending strongly on the kind/density and thickness of the pot) was safely fixed as plane as possible in the sample holder and the food contact side was measured directly by XRF under rotation. The analysis was performed using the energy dispersive X-ray spectrometer Spectro XEPOS-P (Spectro A.I. Ametek, Kleve, Germany). The applied method allows four single measurements per sample using different spectrometer parameters and X-ray tube conditions to optimize the excitation of the sample. To avoid scattering and reabsorption of the fluorescence quants, the measurement was carried out under vacuum condition. This is to improve the sensitivity of the instrument, as the used SDDs (silicon drift detectors) has to resolve all the incoming signals simultaneously. The Spectro XEPOS can resolve, at maximum, ~1 million counts per second. In the first and second set of parameters, 150 s at an energy level from 6 to 19 keV and 150 s at an energy level of above 19 keV were measured, where medium heavy elements and most of the metals are detected. For 150 s at an energy level between 3 and 6 keV and finally 600 s for trace analysis in the energy region from 0 to 3 keV the lighter elements are registered.

When all intensities are captured, the software calculates a spectrum and, after deconvolving the recorded data, the elemental concentrations under consideration of possible overlapping and matrix effects. Thereby the intensity is fitted using a general and standardless calibration model. An overview and semi-quantification of the elemental content in the sample is thus reported within about 15 to 20 min.

## 3. Results

In the present study, styrene migration levels in yogurt and dairy products were measured using purge-and-trap (P&T) in combination with gas chromatography (GC). This technique eliminates the need for organic solvent extraction and laborious cleanup steps to separate target compounds from complex matrices such as foodstuffs: volatile organic compounds (such as styrene) are purged out of the sample matrix (solid or liquid) by an inert gas stream and collected onto a cooled sorbent trap, where they are concentrated and later introduced by rapid heating into the gas chromatograph (GC)/mass spectrometer (MS) for analysis. Since very low detection limits can be achieved, this is a suitable method for the trace analysis of volatiles, such as styrene, in food.

The main parameters of the calibration curves for styrene, the limits of quantification (LOQ) and detection (LOD) (expressed in µg/kg, i.e., part per billion or ppb) in each investigated food matrices obtained with the used purge & trap gas chromatography method are reported in Table 3. The squares of the correlation coefficients (*R^2^*) were higher than 0.999 for all of the matrices tested within the range of interest. For the foods with lower fat content (3.5–10%), lower detection limits were obtained; the higher levels were obtained for the fatty foods with 24% and 30% fat. In the “reference food matrixes” (plain yogurt and dairy products taken from the market packed in PP or glass) styrene was not detectable; the styrene levels measured in the similar products packed in PS can therefore strongly assumed as migrating from the packaging and not due to its natural presence in the food. The recovery rates of styrene in spiked food matrices which were stored with the investigated samples in the freezer were comparable with the one obtained from freshly spiked foods and were in the range 88–111%. Overall, the results showed that the method is sensitive and accurate, suitable for its intended use.

The measured styrene levels in the PS packaging as well as its migration quantified in the respective food at best before date (for some products also before and after it) are summarized in Table 4. The migrated amount is directly expressed as measured in the food considering the real ratio of the PS surface area of the package to the quantity of the food packed for each pot (see S/F ratio reported in Table 2). Since the packages containing the food samples were stored until BBD of the food, it should be noted that the measured residual levels of monomer styrene may underestimate the initial concentration in unfilled packages.

Since the residual levels of monomer styrene in the different PS pots were different, in order to compare the migration levels between the products, the styrene relative migration was determined. It represents the percentage of substance that migrates from the polymer to the food and was calculated by dividing the mass of styrene recovered in the food by the initial mass determined in the packaging. In Table 5, the relative migration of styrene at BBD and the main characteristics of the investigated products (food and packaging) are summarized. An estimation of the TiO_2_ concentrations (as elemental Titanium) measured in seven selected PS packaging is also reported; the obtained results in this case shall be regarded as semi-quantitative because of the rather little amount of material analyzed.

## 4. Discussion

The levels of residual styrene monomer found in the investigated PS food containers which were taken from the supermarket ranged from 256 to 339 µg/g polymer (Table 4), the overall mean was 300 µg/g. In 1976 [34], the styrene mean concentration found in ten yogurt containers was 632 µg/g polymer. In a survey conducted in 1983 on several foods (including yogurt and cream) and their packaging, residual styrene levels in the polymers were reported in the range between 280 and 1400 µg/g [43]. In more recent publications, levels that are more comparable with the ones found in our study were reported in various food grade non-expanded PS resins [24,46,47]. A likely initial concentration in the packaging material of 500 µg/g polymer was assumed in 2007 to perform an assessment of the contribution of styrene from yogurt pots [48]. During the last five decades, manufacturers of food grade PS employed modern post-curing techniques to reduce the residual styrene levels resulting in about 200 µg/g today. This is thought to be the lowest achievable level since the polymer is susceptible to thermal degradation to a certain extent during processing [49]. Despite the relatively low residual levels in food grade commercial PS, styrene monomer migrates into the packed food. The rate of migration from the packaging to the food is a complex issue, influenced by a number of factors including the physical and chemical properties of the polymer and of the migrant, the manufacturing process of the packaging, the composition of the packed food (mostly fat content and structure) and the storage conditions (time and temperature) [50].

In our study, the migration of styrene was investigated in yogurt and dairy products packed in PS which were taken from the German and Italian markets during 2021. All of the chosen products were analyzed at their best before date (BBD), additionally some samples were investigated approx. 20 days before and 15 days after the BBD. Migration results were evaluated and compared with literature data considering several characteristics of each product, such as the residual levels of styrene in the container, the packaging surface to food volume ratio (S/F), the fat content of the packed food, its production process, the storage time/temperature conditions and the date of analysis.

It should be recognized that according to Regulation (EU) No. 1169/2011, it is required that food labels (except in special cases) indicate an expiry date (“To be consumed by”, “Use by”) or a BBD (“Best before by/Best before the end”). The expiry date is the deadline for consumption of the labelled food, after which it will be deemed unsafe for human consumption even when properly stored. The BBD (or minimum shelf-life date) is the date until when, at the latest, an unopened and correctly stored food item must retain its specific properties, such as taste, smell, color, consistency and nutritional value. The decision to put an expiry date or a BBD in a food must be evaluated product by product by the Food Business Operators (FBO), following an approach based on risk analysis. Normally, the dairy industry uses aseptic filling units for packing the products and a controlled cold chain distribution, thus a BBD marking shall be appropriate. However, according a recent EFSA market study [51], the choice for a “use by” date marks predominates for yogurts (64.3%); thus recognizing dairy products as one of the main food categories contributing to food waste. Misinterpretation by consumers of the meaning of the “use by” and “best before” dates can additionally contribute to household food waste. The European Commission will therefore propose, by the end of 2022, the revision of EU rules on date marking, as part of the Circular Economy Action Plan.

In view of the above, it is appropriate to check for the migration extent from the packaging into dairy products such as yogurt and cream also after their BBD.

Styrene migration levels obtained for the investigated refrigerated dairy products taken from the markets (Table 4) range from 5 to 30 µg/kg food when analyzed at their best before date (BBD) and do not seem to increase greatly fifteen days after it. Migration extent measured twenty days before was found approx. be half of that at the BBD for the yogurts with 3.5% fat content and slightly more as the half for the dairy products with a fat content of 10%. However, it has to be considered that only three and four products were analyzed, respectively, after and before their BBD. From a migration theoretical view, it is expected that styrene concentration increases slowly with increasing storage time and partitioning equilibrium will not be approached under the storage conditions.

In the literature, the migrated value of styrene monomer into yogurt and other dairy products packed in PS is quite variable, ranging from <1 to 240 µg/kg (Table 1). The high variability of the reported concentration is mainly explained with the different characteristics of the analyzed products, such as food and packaging composition, time of analysis and storage condition; this information is partially missing in the majority of the published studies, which results in difficult evaluation and comparison of data. In a recent review paper [52], an overall mean of 91.53 µg styrene per kg food was calculated, however, this was based on only four studies, which were selected by quality assessment and meta-regression analysis. The authors further reported that most studies analyzed dairy products and that styrene levels in food were found to be higher, correlating with higher fat content and longer storage time.

The results from our investigation for the refrigerated dairy products (with fat content from 0.1 to 30%) are lower as the overall mean of 90 µg/kg reported in different food [52], but they are in close agreement with recent published data on styrene concentration in yogurt desserts and cream packed in PS containers which were taken from the Greek market in 2020 [40], where levels ranging from <1 to 46 µg/kg depending on days before expiry were reported; however, the study did not specify the fat content of the chosen products and if they were plain (natural) or not. In a survey study from 1984 [41], 146 samples from Victoria and New South Wales, Australia, which included yogurt and cream, the highest level of styrene found was 100 µg/kg in yogurt (without specifying if plain or not and the date of analysis); however, about 85% of all yogurt samples were found to have values less than 50 µg/kg. Data from 1978 [34] for plain yogurt (analyzed after 36 days from packaging) and for sour cream (27 days after packaging) which were packed in PS were reported equal to 13 ± 4 µg/kg and 24 ± 18 µg/kg, respectively, thus in agreement with more recent investigations. Several studies evaluated the effect of storage time on the styrene migration in dairy products and in nearly every case, a progressive increase in the migration with increasing time was found, also depending on the fat content of the products [35,37,40]. However, investigations were performed during the shelf-life of each food and a lack of information still remains on the styrene levels in dairy products after their expiry.

Significantly higher levels, as reported in the literature, were detected in our study for the coffee creamers (UHT products, single portion). Styrene migration in this product, which is stored at room temperature, reached 401 µg/kg when analyzed at the BBD, was already 382 µg/kg twenty days before it and increased up to 465 µg/kg when analyzed fifteen days after it. The BBD was printed on each product and, according to information provided by the manufacturers [53], corresponds to approx. 40 days for the refrigerated yogurt and cream products and to approx. 192 days for the UHT coffee creamers (stored at room temperature). Therefore, on the one hand, the higher styrene levels found in the UHT creamers can be explained by their longer storage time and higher storage temperature conditions, which were much more unfavorable compared to the refrigerated products. It is also important to highlight that the levels of residual styrene monomer found in the packaging of the UHT coffee creamers were the highest among the analyzed containers and these were single serve portions, characterized by a high ratio between the PS surface contact area of the package (0.28 dm^2^) and the quantity of the food packed (10 g). For this product, this surface to volume ratio (S/F) was equal to 28 dm^2^/kg, while for the other investigates products (pots and cups with a food net content between 115 and 200 g sampled from the German and Italian markets) ranged between 8.6 and 11.4 dm^2^/kg. In a recent study [40], the highest concentrations of styrene for dairy products taken from the Greek market were found in small individual portions of milk having higher S/F (equal to 15 dm^2^/kg), compared to the rest of the investigated samples, with S/F values lower than 11 dm^2^/kg. Reported levels for the milk portions from the Greek study range from 76 to 135 µg/kg, depending on time of analysis (days before expiry), which are significantly lower as found for the coffee creamers analyzed in our study; however, the authors reported in [40] that the milk portions were stored at 5 °C and had a shelf-life of approx. 5 months, whereas the coffee creamers analyzed in our study were UHT products stored at room temperature for approx. 6 months. Both the prolonged storage times and the high S/F which were the main characteristic of both these small portions of dairy foods analyzed in our study and, in [40], are suggested to be the main factors influencing the migration of styrene in the packed food. In the case of the UHT products we analyzed, the additional factor which contributed to an increase in the migration extent was the temperature.

According to the EU cube model, the standard surface to volume ratio for food packaging (calculated dividing the contact surface areas of the packaging by the mass of food) is considered 6 dm^2^/kg of food. This conversion value is applied to check the compliance with the relevant migration limits of plastic food packages that have a volume lower than 500 milliliters or grams or more than 10 L (except for those intended to come into contact with food for infants and young children). In the case of the PS pots for dairy products from the market, the real S/F ratios are slightly higher (in the case of the refrigerated products) or even four times higher (in the case of single serve portions) compared to this conventional value. As previously described [54], the influence of the S/F on the migration extent depends on the solubility of the substance in the simulant or the food; there are two borderline cases: (1) the substance is a good soluble in the food or simulant, i.e., the partitioning equilibrium is on the side of the food or simulant, then the increase in the surface to volume ratio does not influence the area-related migration; (2) the substance is insoluble in the food or simulant, i.e., the partitioning equilibrium is strongly on the side of the polymer, then the concentration in the food or simulant is independent from the simulant/food volume and depends on the concentration in the polymer and the partition coefficient only. With regards to styrene, it is soluble in alcohol and organic solvent, slightly soluble in water (300 mg/L at 20 °C) and has an octanol/water partition coefficient of 2.95, indicating higher solubility in fatty (nonpolar) media; therefore the effect of the S/F on the styrene migration extent could depend on the food product composition and play an important role if considering fatty foods.

Looking more in detail at the results obtained in our study for the level of migrated styrene in yogurt analyzed at their BBD, it can be observed that higher values were found in the set products with a fat content ranging from 3.5% to 3.9% and average migration of 24.3 µg/kg (item 4-5-6-7 Germany) compared with the stirred products with 3.5% fat content which resulted in an average migration of 11.3 µg/kg (item 1-2-3 Germany). According to the information provided by the dairy producers [53], the analyzed stirred and set yogurt are produced in different ways. For the stirred yogurt, the inoculated milk is fermented and stirred in large stainless steel fermentation tanks, after few hours is pre-cooled at a temperature of 18–20 °C and filled in the packaging (i.e., PS pots) where it is allowed to cool down to 5 °C for the distribution. In contrast, during the production of the set products, the milk is fermented in the retail packaging at 46 °C, cooled at 44 °C within 5-7 h or until the desired acidity (pH <4.4) is attained, only afterwards cooled down to 5 °C for distribution. Both products have a shelf life of approx. 40 days. Therefore, the temperature of contact between the PS containers and the yogurt does not exceed 20 °C in the first hours after packaging and remain below 5 °C during shelf life in the case of the stirred yogurt; contrasting the higher temperature of approx. 40 °C for the first 8 h during the production of the set-products. The average styrene levels found in our study for the set yogurts were higher compared to the stirred yogurts, thus suggesting that the incubation at 40 °C also has a significant impact on the styrene migration and can be considered as the main reason for the higher styrene levels found in the set yogurts.

In many European countries, including Italy, the stirred yogurts are preferred and are also often sold sweetened, as light or creamy products. The Migration of styrene in yogurt samples taken from the Italian market was found lower (approx. the half) in a light product with 0.1% fat (item 3 Italy: 6 µg/kg) compared to a creamy one with 4.4% fat (1 Italy: 18 µg/kg). Once more, these results, albeit limited to a few products, seem to confirm the relation between styrene migration and fat content of the food as reported in previous studies.

In the other refrigerated dairy products, such as whipping and sour cream (item: 8-9-10-11-12-13 Germany), styrene migration measured at their BBD ranged from 5 to 18 µg/kg. These values are again in close agreement with the one recent published for cream products taken from the retail market in Greece [40], whereby the concentrations ranged from 5 to 16 µg/kg, depending on the time of analysis from expiry of the products. Interestingly, the average styrene level found in our study for cream products (12 µg/kg) was comparable with the one found in the stirred yogurts with 3.5% fat content (item 1-2-3 Germany: 11 µg/kg) and lower as the set yogurts with 3.5–3.9% fat content (item 4-5-6-7 Germany: 24 µg/kg). It is worth noting that, according to the information provided by the producers, fermented cream products (sour creams) can also be produced in different way; stirred sour cream, as stirred yogurt, is fermented in tanks and filled in the PS pots after cooling to approx. 18–20 °C; while fat or semi-solid sour cream is fermented in the PS packaging (approx. at 30 °C for 2–3 h), but the conditions vary between different producers, which does not allow a clear distinction of the products such as for yogurt.

It has to be highlighted that the residual content of styrene monomer determined in the investigated PS pots (*C_p,0_*) differed. No clear relationship between the styrene monomer content of the package and the migrated amount into the foodstuffs (comparable for fatty content and production process) emerged from our study (see Table 4). The main explanation for this finding could be that the variation in the residual monomer content in the packaging was rather small, as expected because polymer manufacturers tent to reduce this in food grade PS. However, in order to better evaluate the migration levels in the different food products and understand the effect of the production type, packaging characteristics and fat content of the food the relative migration (percentage of substance that migrates from the polymer to the food) for each product was calculated (see Table 5). Looking at the obtained values, it can be seen that the relative migration of styrene among the investigated refrigerated dairy products ranged from 0.04% to 0.29% of the total residual content of monomer in the PS pots, while in the case of the UHT product stored at RT, it reached 1.66%. In the stirred yogurts with 3.5% fat content (item 1-2-3 Germany) lower relative styrene migration values (average: 0.14%) were determined compared to the set yogurts with 3.5–3.9 % fat content (item 4-5-6-7 Germany), for which an average relative migration of 0.22% was calculated. Among the Italian stirred yogurts for the light product (0.04% fat), the lowest relative migration of styrene (0.04%) was observed; in the creamy one relative migration reached 0.14%. The average relative styrene migration into the refrigerated cream products was 0.16%, which is comparable with the one found in the stirred yogurts with 3.5% fat content. A correlation between the fat content and the relative styrene migration levels in the different cream products (with 10% to approx. 30% fat content) was not found. However, it has to be noted that only one sour cream product with 24% fat and four products with 10% fat were investigated. The whipping cream (30% fat) was not homogeneous (cream floating at the top, only minimally in contact with the PS pot) which complicates comparisons.

Additionally, seven PS packaging (item 1 to 7 Germany)were analyzed with regards to their titan dioxide (TiO_2_) content. TiO_2_ is widely added to polystyrene for food contact applications in order to improve the appearance of the package and to provide some barrier to light. According to the literature [55], the amount of TiO_2_ migrated to packaged food is negligible. Interestingly, a study [23] showed that organoclay and zinc oxide nanoparticles (ZnO) could affect (decrease) styrene migration from PS into food simulants. Data on the effect / influence of TiO_2_ on migration of monomers and other additives from plastic into food are not currently available. The purpose of our analyses was to investigate the correlation between the TiO_2_ content in the PS pots and the measured migration levels of styrene in yogurt. It was observed that six out of seven investigated PS pots contained TiO_2_, and the concentration varied greatly (from approx. 2.3% to 0.01% *wt*/*wt*). A correlation between the TiO_2_ content and the relative styrene migration levels (see Table 5) could not be identified. It has to be considered that only seven samples were analyzed and that the methodology used could not distinguish if nano-TiO_2_ was used. Even more importantly, the specific composition of the PS materials was unknown; it is hypothesized that the polymer blends constituting the packaging and the additives used were not the same, thus the migration extent was additionally influenced by the different diffusive properties of the materials. This therefore makes comparing the products from the market difficult regarding the influence of TiO_2,_ which is expected to be relatively small. To determine the effect of the TiO_2_ content in PS and the migration levels of styrene in simulant/foods, further research work is needed, using identical PS materials with several known added concentrations of TiO_2_.

Clearly, many complex factors affect the relative migration rate and the total amount of styrene monomer which passes into the foodstuffs. We found in our investigation on real dairy products taken from the market that the ratio between the PS surface contact area of the package and the quantity of the food packed (S/F), the time/temperature conditions of production/packaging filling as well as of storage/shelf life of the products are the main factors influencing migration of styrene monomer into real foods taken from the market.

With the limited data available to date, it can be shown that actual migration of styrene into refrigerated dairy products (real products taken from the market) is generally much lower as obtained in the corresponding food simulant (50% ethanol) under standardized condition. Indeed, migration from HIPS was recently investigated, testing extruded sheets by total immersion under the standardized conditions 10 days/40 °C (which are laid down in the EU legislation to simulate all storage times at refrigerated and frozen conditions including hot-fill) with the food simulants 50% ethanol (simulant for dairy products) [27]. Styrene migration under these conditions reached 297 µg/kg in 50% ethanol (calculated applying a surface/volume ratio of 6 dm^2^/kg), which corresponded to a relative migration of approx. 7%. Swelling (uptake of liquid by the PS polymer due to interactions with ethanol) reached 1.3%, which was considered the cause of exaggerated migration values. Within the same study, styrene levels of 152 µg/kg (3.6% relative migration) and 278 µg/kg (6.6% relative migration) were determined, respectively, in milk (3.5% fat) and cream (30% fat) tested by total immersion at 10 days/40 °C. Swelling with real foods was not observed. Migration in these foods, tested at 40 °C by total immersion, was much higher than in the refrigerated milk products taken from the market which were investigated within the present study. The higher temperature increases the migration rate strongly; the test specimen edges in contact with the media (food or food simulant) might also affect the styrene migration behavior. The highest styrene migration level found with the present study in the refrigerated dairy products (yogurts and creams packed in PS) taken from the market was 30 µg/kg, which is approx. ten times lower as found migrating from HIPS sheets in 50% ethanol, tested at 10 days/40 °C by total immersion. On the other hand, styrene levels in the coffee creamers in PS single portions which were UHT products stored at room temperature for approx. 6 months with a surface to volume ratio (S/F) of 28 dm^2^/kg were determined at BBD equal to 401 µg/kg. In our previous study [27], styrene migration from HIPS sheets was determined in 50% ethanol tested by total immersion at 10 days/60 °C (prescribed time/temperature condition in the EU legislation to simulate contact times longer than 30 days at room temperature and below) equal to 341 µg/kg, calculated applying a surface/volume ratio of 6 dm^2^/kg (according to the EU cube model). This concentration of styrene can be recalculated using a S/F ratio of 28 dm^2^/kg (for simulating the application: single serving coffee creamer) instead of the conventional ratio of 6 dm^2^/kg equal to 1591 µg/kg.

In the light of the above mentioned, it is possible to conclude that migration tests applied within laboratory simulations with 50% ethanol to evaluate regulatory compliance appear too severe and are not representative for the majority of the real food packaging application of PS, which are mostly refrigerated dairy products (such as yogurt and cream) with single side contact and a shelf-life of approx. 40 days.

Additionally, it has to be considered that the S/F ratio values of dairy products packed in PS mainly differs from the EU standard ratio of 6 dm^2^/kg, especially in the case of small plastic packages (such as single serving applications). This conventional S/F ratio shall compensate for the smaller amount of food consumed from such small packages. This is also worth considering in the context of health risk assessment. Indeed, the default (worst-case) assumption in Europe is that every day an adult person consumes 1 kg of food packaged in a 1 dm^3^ cube with a surface area of 6 dm^2^; this is a conservative assumption that does not reflect the real consumption patterns. Indeed, the real exposure of consumers to styrene via the consumption of foods packed in small packages will be reasonably lower. For instance, the styrene levels found in the coffee creamers packed in PS single portions (each containing only 10 g of food) were equal to 401 µg/kg at BBD; the corresponding real exposure of the consumer in this case would be therefore only 4 µg per portion.

If the food and packaging industry as well as the authorities are to continue to use simulants migration testing giving much higher levels of migration than real food migration, then a pragmatic way forward will be required. Such a way forward could involve the generation of data on styrene migration into food to see if a robust correlation between migration into recommended simulants/under standard conditions and real food migration can be determined, justifying the use of a reduction factor (similar to the approach for fatty foods already included in the food packaging legislation).

Further research work is envisaged which should cover the main product categories with PS packaging available on the market as well as worst case conditions of use (such as small packaging, stored at room temperature or hot fill contact applications) in order to gain more knowledge on styrene migration into real foods and to guarantee consumer safety.

## 5. Conclusions

In this study, the migration of styrene was investigated in yogurt and dairy products packed in PS which were taken from the German and Italian market during 2021 using purge-and-trap (P&T) in combination with gas chromatography (GC). The used method showed low detection limits (0.5 µg/kg for yogurt with 3.5% fat content and 3 µg/kg for cream with 30% fat content) and can be considered suitable for trace analysis of volatiles, such as styrene, in food.

Styrene migration levels in the refrigerated dairy products taken from the market (PS pots with net content ranging from 115 to 200 g) ranged from 5 to 30 µg/kg food when analyzed at the end of their shelf life (approx. 40 days). This is in close agreement with recently published data on styrene concentration in yogurt desserts and cream packed in PS containers which were taken from the Greek market. Significantly higher levels as reported in the literature were detected in the present study for UHT coffee creamers (10 g single portions). Styrene migration in this room temperature stored product, reached 401 µg/kg when analyzed at the end of their shelf life (approx. 6 months), which corresponds to an exposure of 4 µg styrene per single portion. Among the several parameters evaluated, the ratio between the surface contact area of the package and the quantity of the food packed (S/F), the time/temperature conditions of production/filling as well as of storage of the products were identified as the main factors influencing styrene migration into food under real application of use. The fat content might influence the styrene level but the effect, if any, was too small in relation to the variability of other parameters.

Actual migration of styrene into refrigerated food packaged in PS was found to be much lower than that measured migrating into 50% ethanol food simulant under standardized condition. This raises the question as to whether for PS plastics compliance testing should be adapted taking into consideration the correlation between migration testing by laboratory simulations and migration in real food. Further research work is envisaged in order to gain more knowledge on styrene migration into real foods and to allow the conversion of simulated migration test results in a more realistic migration scenario also considering the corresponding real exposure of the consumer.

## Figures and Tables

**Table 1 foods-11-02120-t001:** Levels of styrene (µg/kg) in yogurt and other dairy products packed in PS from retail reported in literature as mean or range values.

Yogurt ^1^	Sour Cream	Butter/Margarine ^2^/ Fat Cream	Reference
36–44			[14]
2–9			[30]
4–240			[31]
4		8–9	[32]
4			[33]
3–35			[34]
9–78	4–246	22–59	[35]
<1–16			[36,37]
8–12		49–51	[38]
11	28	74–102	[39]
1–46		5–16	[40]
<1–100	<1–5	<1–20	[41]
19		2–18	[42] ^3^
<1–200		<1–70	[43]

^1^ Including plain (natural) and also flavored products, with different fat content. ^2^ It should be mentioned that margarine containers are likely not made of PS but of acrylonitrile-butadiene-styrene (ABS) resins. ^3^ Food samples (composites) were collected without taking account of the packaging material.

**Table 2 foods-11-02120-t002:** Description and main characteristics of the investigated products (food and packaging).

Item/Country	Food Description	Fat Content [%]	Net Content [g]	Packaging Description	Area Weight [g dm^−2^]	S/F ^1^ [dm^2^ kg^−1^]
Refrigerated products (Shelf life: 40 days/5 °C)						
1 Italy	Stirred, creamy yogurt	4.4	125	PS pot-No labels/no printing	3.58	11.0
2 Italy	Stirred, sugared yogurt	3.2	125	PS pot-Outside printed	3.29	11.4
3 Italy	Stirred, light yogurt	0.1	125	PS pot-No labels/no printing	3.78	10.6
1 Germany	Stirred yogurt	3.5	150	PS pot-Bonded paper label	3.43 ^2^	9.3
2 Germany		3.5	200	PS pot-Removable paper-banderole	2.02	8.6
3 Germany		3.5	115	PS pot-Bonded paper label	3.81 ^2^	10.7
4 Germany	Set yogurt (fermented in the pot)	3.8	200	PS pot-Bonded plastic label	4.40 ^2^	9.9
5 Germany		3.9	150	PS pot-Outside printed	3.35	9.9
6 Germany		3.5	150	PS pot-Outside printed	3.83	10.3
7 Germany		3.5	150	PS pot-No labels/no printing	3.43	9.1
8 Germany	Whipping cream (fluid, with cream on top)	30 ^3^	200	PS pot-Removable paper-banderole	2.27	9.8
9 Germany	Fat sour cream (solid)	24	200	PS pot-Outside printed	3.65	9.9
10 Germany	Sour cream (solid)	10	200	PS pot-Removable paper-banderole	2.29	9.4
11 Germany	Sour cream (solid)	10	200	PS pot-Outside printed	3.39	10
12 Germany	Sour cream stirred (semi-solid)	10	200	PS pot-Outside printed	3.99	9.4
13 Germany	Sour cream stirred (semi-solid)	10	200	PS pot-Removable paper-banderole	2.02	9.4
UHT products (Shelf life: 192 days/RT)						
14 Germany	Coffee creamers	10	10	PS pot multipack. No labels/no printing	2.55	28

^1^ S/F: Ratio of the PS surface area of the package to the quantity of the food packed. ^2^ Including label. ^3^ The specification 30% fat does not represent a homogenous value for the whole product (cream floating at the top, only minimally in contact with the PS pot).

**Table 3 foods-11-02120-t003:** Main parameters of the calibration curves for styrene, limits of quantification (LOQ) and detection (LOD) in the investigated food matrices.

Matrix	Range [µg/kg]	*R* ^2^	Recovery ^1^ [%]	LOQ [µg/kg]	LOD [µg/kg]
Yogurt 3.5% fat	4–61	0.9999	98	1.7	0.5
Cream 30% fat	5–61	0.9992	98	10.0	3.0
Sour cream 24% fat	12–61	0.9996	101	3.8	1.1
Sour cream 10% fat	5–49	0.9998	88	2.7	0.8
Coffee cream 10% fat	12–838	0.9993	111	1.2	0.3

^1^ Recovery calculated as measured concentration/spike level.

**Table 4 foods-11-02120-t004:** Residual levels (*C_p,0_*) in the PS packaging and migrated amount of styrene in food before/at/after best before date (BBD).

Item/Country	Food Description	Styrene *C_p,0_* [µg/g]	Migration before BBD ^1^ [µg/kg]	Migration at the BBD [µg/kg]	Migration after BBD ^2^ [µg/kg]
Refrigerated products (Shelf life: 40 days/5 °C)					
1 Italy	Stirred yogurt, 4.4% fat	316 ± 8		18 ± 0.1	
2 Italy	Stirred yogurt, 3.2% fat	266 ± 1		13 ± 0.2	
3 Italy	Stirred yogurt, 0.1% fat	356 ± 2		6 ± 0.1	
1 Germany	Stirred yogurt, 3.5% fat	351 ± 23		17 ± 0.1	26 ± 0.2
2 Germany	Stirred yogurt, 3.5% fat	275 ± 2	5 ± 0.1	10 ± 0.2	
3 Germany	Stirred yogurt, 3.5% fat	284 ± 5		7 ± 0.2	9 ± 0.1
4 Germany	Set yogurt, 3.8% fat	357 ± 14		27 ± 0.5	
5 Germany	Set yogurt, 3.9% fat	256 ± 18		25 ± 0.2	
6 Germany	Set yogurt, 3.5% fat	278 ± 12	7 ± 0.2	15 ± 0.2	
7 Germany	Set yogurt, 3.5% fat	308 ± 6		30 ± 0.4	30 ± 0.5
8 Germany	Whipping cream with 30% fat ^3^	265 ± 21		13 ± 0.1	
9 Germany	Fat Sour cream with 24% fat	300 ± 34		18 ± 0.6	
10 Germany	Sour cream with 10% fat	306 ± 60		14 ± 0.6	
11 Germany	Sour cream with 10% fat	296 ± 8	8 ± 0.1	10 ± 0.2	
12 Germany	Stirred Sour cream with 10% fat	292 ± 20		5 ± 0.6	
13 Germany	Stirred Sour cream with 10% fat	260 ± 8	7 ± 0.1	11 ± 0.6	
UHT products (Shelf life: 192 days/RT)					
14 Germany	Coffee creamers with 10% fat	339 ± 56	382 ± 1.8	401 ± 1.2	465 ± 1.9

^1^ Analysis 20 days before BBD of the products. ^2^ Analysis 15 days after BBD of the products. ^3^ The specification 30% fat does not represent a homogenous value for the whole product (cream floating at the top, only minimally in contact with the PS pot).

**Table 5 foods-11-02120-t005:** Relative migration of styrene at BBD and main characteristics of the investigated products (food and packaging).

Item/Country	Food Description	TiO_2_ in PS [µg g^−1^]	Area Weight [g dm^−2^]	Fat Content [%]	S/F ^1^ [dm^2^ kg^−1^]	Relative Migration [%]
Refrigerated products (Shelf life: 40 days/5 °C)						
1 Italy	Stirred, creamy yogurt		3.58	4.4	11.0	0.14
2 Italy	Stirred, sugared yogurt		3.29	3.2	11.4	0.12
3 Italy	Stirred, light yogurt		3.78	0.1	10.6	0.04
1 Germany	Stirred yogurt	5647	3.43 ^2^	3.5	9.3	0.16
2 Germany		23580	2.02	3.5	8.6	0.20
3 Germany		150	3.81 ^2^	3.5	10.7	0.06
4 Germany	Set yogurt (fermented in the pot)	<0.3	4.40 ^2^	3.8	9.9	0.18
5 Germany		20620	3.35 ^2^	3.9	9.9	0.29
6 Germany		13350	3.83 ^2^	3.5	10.3	0.13
7 Germany		3510	3.43	3.5	9.1	0.27
8 Germany	Whipping cream (fluid, with cream on top)		2.27	30 ^3^	9.8	0.20
9 Germany	Fat sour cream (solid)		3.65	24	9.9	0.17
10 Germany	Sour cream (solid)		2.29	10	9.4	0.23
11 Germany	Sour cream (solid)		3.39	10	10	0.12
12 Germany	Sour cream stirred (semi-solid)		3.99	10	9.4	0.05
13 Germany	Sour cream stirred (semi-solid)		2.02	10	9.4	0.23
UHT products (Shelf life: 192 days/RT)						
14 Germany	Coffee creamers		2.55	10	28	1.66

^1^ S/F: Ratio of the PS surface area of the package to the quantity of the food packed. ^2^ Including label. ^3^ The specification 30% fat does not represent a homogenous value for the whole product (cream floating at the top, only minimally in contact with the PS pot).

## Data Availability

Not applicable.

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
