# Peer review of "Migration of Styrene in Yogurt and Dairy Products Packaged in Polystyrene: Results from Market Samples"

_foods, 2022, doi:10.3390/foods11142120_

Round 1

Reviewer 1 Report

This papers studies the presence of styrene in polystyrene based packages and in dairy food packaged in these containers. The method is used is adequate and well described, the results are well discussed by comparison with abundant literature references on this issue. The results and conclusions are also relevant since styrene in food contacting materials is being reevaluated by EFSA. 

Author Response

Many thanks for this positive feedback!

Reviewer 2 Report

The subject dealt with in the manuscript is important and current. My only concern with the manuscript is that the total number of samples is small and the total sample contains many different varieties. On the other hand, the manuscript in general is well organized.

Author Response

Many thanks for this positive feedback! We analyzed seventeen samples and we wrote (see line 151) that indeed this is a limited number. However, we do not think that this is a too small number of samples; in literature, we found some studies with focus on styrene levels in real food which were conducted with even fewer samples. Moreover, we discussed our results by comparison with several literature references.

Reviewer 3 Report

Polystyrene is a non-biodegradable synthetic polymer. It has been extensively used for developing packaging materials. Among the various applications of such packaging materials, packaging dairy products under refrigerated conditions is one of the applications of polystyrene packaging systems. However, it has been reported that these packaging materials may leach styrene. Accordingly, the European Food Safety Authority is re-evaluating styrene to assess the safety of food contact materials.

The authors have sampled 17 dairy products (yogurt and cream) from the Italian and German markets in the present study. The analysis of the refrigerated dairy products indicated the presence of styrene at concentrations of 5-30 µg/g. The authors also found that each serving portion of coffee creamer consisted of 401 µg/g after 190 days of storage at room temperature. The styrene migration depended on the surface contact area of the packaging material, quantity of the food packed, packing conditions, and storage conditions. The authors also found that the migration of styrene was slower when yogurt was fermented in an incubator and filled at 20 oC than when it was fermented at higher temperatures. The fat content in the food products had a negligible effect on the styrene migration. Further, the authors reported that food simulants showed much higher levels of styrene migration.

The following are my observations:

1.       The use of polystyrene-based packaging systems for food products is extensive. Understanding the mechanics of styrene migration can help to understand its diffusion kinetics. I would be excited if the authors could include the migration kinetics of styrene molecules.

2.       I would suggest performing 2D-FTIR spectroscopy or 2D Raman spectroscopy to understand the distribution of the styrene molecules.

3.       The quality of the English language is okay

4.       Adequate references have been provided to justify the results

Author Response

Many thanks for this positive feedback and suggestions! In response to your observations: we plan further research work on food packaging made of polystyrene. We would like to perform some spectroscopy analyses to understand the material composition and also to investigate the migration kinetics of styrene in the near future. However, the present manuscript focuses on styrene levels in real food and we think we discussed obtained results with relevant literature references.